# OpenReview forum: "SafeSpec: Fast and Safe LLM via Dynamic Reflective Sampling"
_ICML.cc/2026/Conference — ICML 2026 regular_

### Official Review · Reviewer_E4Gv · 2026-03-09

**Soundness:** 3
**Presentation:** 3
**Significance:** 2
**Originality:** 3
**Overall Recommendation:** 4
**Confidence:** 3

**Summary:**

The paper propose a method called SafeSpec for fast and safe speculative decoding. They propose to calculate a quality score and a safety score (given by a safety head trained with carefully collected samples) to verify the answer of the smaller model. For unsafe candidates, to avoid over-defense issue, they use multi-sampling and rollback with Self-Reflection to regenerate more safer candidates. The experiments show that the three components all contribute to the overall framework.

**Compliance With Llm Reviewing Policy:**

Affirmed.

**Final Justification:**

The rebuttal addresses many of my concerns to some extent on experiments, including the sensitivity analysis of reflection prompts and the discussion of the related works. However, I think the authors should be more careful with the table presentation in the revision. Overall, I think the author did a good job in combining safety and speculative decoding, which is somewhat innovative. Therefore, I recommend weak acceptance.

**Key Questions For Authors:**

1. For the self-reflection, how do you choose the one as shown in Figure 3? Are there alternatives?

2. In figure 4(c), which dataset are you using? I am wondering whether the trend would be different for different datasets with different levels of difficulties. If so, then does that mean the threshold needs to be carefully determined for every new dataset?

3. How long does it take to train the safety head?

4. please also consider to discuss some of the following related work:

[1] Speculative Safety-Aware Decoding

[2] Root Defense Strategies: Ensuring Safety of LLM at the Decoding Level

**Limitations:**

yes

**Strengths And Weaknesses:**

**Strength**
- The idea of combining speculative decoding and safety is valuable since we dont need to retrain the model.
-  The writing is clear and easy to follow. The diagram is also clear.

**Weakness**
- I am concerned with the performance of SafeSpec. The authors bold the value of their own method instead of the best. I think it might be misleading, and there are no explanations of what bold means throughout the paper.
- Lack of discussion of the sensitivity of the reflection prompt.
- Since the SafeSpec applies multi-sampling, Spec-Decoding+multi-sampling and Specreason+multi-sampling should be baselines as well.

---

> ### Author Rebuttal · Authors · 2026-03-29
>
> We sincerely thank you for the constructive feedback. We address each concern below.
>
> ---
> ## W1: Bold formatting in tables
>
> Thank you for point this out. Bold entries currently indicate SafeSpec's results for easy identification, not the best per column. We will clarify this convention and highlight the best result in each column in the revision.
>
> ---
> ## W2 & Q1: Reflection prompt sensitivity
>
> We thank you for raising this important point. To select the reflection prompt used in our paper (Prompt B, shown in Figure 3), we tested several prompts of varying strictness on a small validation set and chose the one that best balanced ASR and over-refusal (XSTest). To evaluate sensitivity, we further compared four prompts ranging from mild to strict, where Prompt C is a paraphrase of B with different wording ([full results here](https://anonymous.4open.science/api/repo/Fig-8B4E/file/reflection_prompt_sensitivity.png)). Overly mild prompts (A) lead to higher ASR, while overly strict prompts (D) cause excessive over-refusal. Prompts B and C perform consistently well, indicating robustness to prompt wording.
>
> We will include this analysis in the revision.
>
> ---
> ## W3: Multi-sampling baselines
>
> Multi-sampling requires a safety judge to select among candidates. Spec-Decoding and Specreason lack such a judge, so there is no criterion to decide which candidate to keep. Therefore, these baselines are non-trivial to define.
>
> ---
> ## Q2: Dataset for Figure 4(c) and threshold sensitivity
>
> All subfigures in Figure 4 use the same datasets: ASR averages 7 jailbreak methods, XSTest measures over-refusal, and ACC/Speed use three benign benchmarks (GSM8K, MATH, GPQA).
>
> To further address whether $\tau_q$ needs per-dataset tuning, we provide a per-benchmark breakdown:
>
> | $\tau_q$ | GSM8K (↑) | MATH (↑) | GPQA (↑) |
> |:--------:|:-----:|:----:|:----:|
> | 6        | 0.90  | 0.64 | 0.43 |
> | 7        | 0.92  | 0.64 | 0.45 |
> | 8        | 0.96  | 0.66 | 0.42 |
> | 9        | 0.96  | 0.78 | 0.52 |
>
> The quality score ranges from 0 to 9. From our experiments, we observe that harder benchmarks are more sensitive to $\tau_q$ (from 9 to 6: GSM8K −6%, MATH −14%, GPQA −9%). However, our results also show that **$\tau_q$ does not require per-dataset tuning**: by setting $\tau_q = 9$ (the maximum, as used in our main experiments), quality is prioritized, which achieves minimal utility loss while maintaining high efficiency across all difficulty levels without calibration.
>
> ---
> ## Q3: Safety head training time
>
> The safety head is a two-layer MLP (10M–30M params) trained with the backbone frozen. For Qwen3-32B, training on ~10K samples takes ~2 hours on 2×A800 GPUs. For DeepSeek-70B, the safety head requires ~4 hours on 4×A800 GPUs.
>
> ---
> ## Q4: Related work — SSD and RDS
>
> Thank you for highlighting these concurrent works. While both address the speed-safety trade-off in LLMs, they differ from SafeSpec in key ways.
>
> - SSD [1] requires a safety-aligned draft model, whose output distribution is mixed with the target model's to quantify jailbreak risk. This imposes an alignment tax on the draft model, degrading utility (see our Q1 response to Reviewer wfmP). Furthermore, safety signals derived from a small model's output distribution are inherently less reliable than probing the target model's internal representations. SafeSpec instead enforces safety entirely at the target model level via hidden state probing, requiring no modification to the draft model.
> - RDS [2] introduces an EAGLE-style head to approximate target hidden states for per-token harmfulness classification, requiring an additional predictive module trained separately for each target model. SafeSpec directly reads the target model's actual hidden states at segment boundaries through a lightweight MLP, avoiding any approximation overhead.
>
> RDS's released code lacks key components for reproduction, so we compare against SSD on Qwen3-32B (SSD uses Qwen3-4B-SafeRL as draft; SafeSpec uses Qwen3-4B):
>
> | Method | AVG ASR (↓) | XSTest (↓) | AVG ACC (↑) | Speedup (↑) |
> | --- | --- | --- | --- | --- |
> | SSD | 0.13 | 0.05 | 0.77 | 1.21× |
> | SafeSpec | 0.05 | 0.11 | 0.76 | 1.95× |
>
> As shown in the table, SafeSpec achieves **lower ASR** with **higher speedup (1.95× vs 1.21×)**, highlighting the effectiveness of our method.
>
> We will include a full discussion in the revision.
>
> [1] Wang et al., *Speculative Safety-Aware Decoding*, EMNLP 2025.
>
> [2] Zeng et al., *Root Defence Strategies: Ensuring Safety of LLM at the Decoding Level*, ACL 2025.

---

> > ### Author Rebuttal · Reviewer_E4Gv · 2026-04-01
> >
> > Thanks for the authors' response. They address all of my concerns, and I will raise the score accordingly.

---

> > > ### Author Response · Authors · 2026-04-01
> > >
> > > We sincerely thank you for the increased score and for the constructive feedback throughout the review process. Your insightful comments have helped us improve the paper, and we are grateful for the time and effort you have invested in evaluating our work.

---

### Official Review · Reviewer_L9Se · 2026-03-10

**Soundness:** 3
**Presentation:** 3
**Significance:** 3
**Originality:** 3
**Overall Recommendation:** 5
**Confidence:** 4

**Summary:**

This paper proposes SafeSpec, a safety-aware speculative inference framework that integrates risk estimation directly into the verification process. SafeSpec attaches a lightweight latent safety head to the target model to jointly evaluate semantic validity and safety in a single forward pass. When unsafe generations are detected, SafeSpec applies rollback and safety-guided reflective multi-sampling to recover safe continuations rather than terminating generation. We model jailbreak attacks as distributional shifts over generative trajectories, where adversarial prompts increase the probability of harmful continuations without eliminating safe ones. Under this model, SafeSpec performs risk-aware trajectory recovery within the speculative decoding process. Across multiple models and adversarial benchmarks, SafeSpec achieves a substantially improved safety–efficiency trade-off. On Qwen3-32B, SafeSpec reduces attack success rates by 15% while preserving a 2.06× inference speedup, demonstrating that speculative acceleration and inference-time safety can be jointly optimized.

**Compliance With Llm Reviewing Policy:**

Affirmed.

**Final Justification:**

The authors have addressed all my concerns. I maintain my positive score.

**Key Questions For Authors:**

# Questions
1. What's the guidance for choosing a draft model in practice. Will the performace be better if the draft model is SFT/RL trained to be the aligned one, or direcly use a better aligned model?
2. There are nunmerous attack baselines for jailbreak attacks. I know it's impossible to cover all of them. However, based on my experience, the flipattack is an effective jailbreak technique. Can you provide the defense performance of SafeSpec against the FlipAttack?
FlipAttack: Jailbreak LLMs via Flipping; Yue Liu, Xiaoxin He, Miao Xiong, Jinlan Fu, Shumin Deng, Bryan Hooi; ICML 2025

**Limitations:**

The authors did not adequately discussed the limitations. I have proposed several suggestions in weakness 2, question 1-2 for more detailed training settings and guidelines.

**Strengths And Weaknesses:**

# Strength
1. This paper focus on the novel issue that the existing defense methods are not designed for the speculative inference. Directly applying the existing defense methods to the speculative inference will lead to the performance degradation in both breaking the draft-verify mechnism or degrading the speed. This paper propose a new framework to address this issue and it's a novel contribution.
2. The experiments are comprehensive and convincing. They cover multiple models and adversarial benchmarks. Especially, they consdier the larger model than the most of the previous works, and use XSTEST to evluate the overrefual performance.
3. The detailed design enables the low overhead for the speculative inference. They reuses the hidden states that are already computed for the generation, and only add a small overhead for the safety evaluation. Morever, they design a Safety-Guided Multi-Sampling and Recovery to avoid dropping all the content when the unsafe generation is detected. Overall, I think the idea is novel and the design is solid.

# Weakness
1. The detection is based on safe head. While the design ensure low computational overhead, the generalization ability is not clear as the head is trained on jailbreak attacks with the similar structure. Could you provide more experiments to evaluate the generalization ability by providing the precision/recall/f1 at different thresholds and the attack out of the domain (e.g., multilingual jailbreaks)?
2. While the Figure 1 is attractive, the author did not explain how the specifiac number is obtained, especially which dataset is used to evaluate the latency performance. I believe that the latancy should be more rigorous to be compared on different datasets. For example, 1) the pure benign dataset; 2) jailbreak dataset; 3) XSTEST dataset, 4) mixed dataset, etc.

---

> ### Author Rebuttal · Authors · 2026-03-29
>
> We sincerely thank you for recognizing the novelty and solidity of our design. We address each of your concerns below.
>
> ---
> ## W1: Safety head precision/recall/F1 and generalization
>
> Thank you for the suggestion. We evaluate the safety head's step-level classification under three settings: **ID** (validation split), **OOD** (6 unseen jailbreak methods: ABJ, AttntionShift, CodeChameleon, ReNeLLM, Mousetrap, and H-CoT), and **Multilingual** (multilingual jailbreak prompts [1]).
>
> |$\tau_s$|Prec. (ID / OOD / Multi)|Rec. (ID / OOD / Multi)|F1 (ID / OOD / Multi)|
> |---|:---:|:---:|:---:|
> |0.2|0.975 / 0.978 / 0.927|0.834 / 0.804 / 0.479|0.899 / 0.882 / 0.632|
> |0.4|0.940 / 0.970 / 0.812|0.919 / 0.871 / 0.879|0.929 / 0.918 / 0.844|
> |0.6|0.897 / 0.953 / 0.755|0.956 / 0.907 / 0.950|0.926 / 0.930 / 0.841|
>
> | Method | ASR (Multi. ↓) |
> |:---|:---:|
> | No Defense | 0.10 |
> | SafeSpec | 0.00 |
>
> **OOD generalization is strong.** F1 degrades by less than 1% from ID to OOD across all thresholds, confirming the safety head learns generalizable features rather than overfitting to specific attack patterns.
>
> **Multilingual gap is moderate and expected.** F1 drops to ~0.84, consistent with findings that alignment-induced separation is weaker for non-English languages in the latent space [2]. Since the safety head operates on last-layer hidden states where language-specific features are most prominent [3], the decision boundary does not fully transfer cross-lingually. Despite this, SafeSpec still reduces multilingual ASR from 10% to 0%, demonstrating robustness under distribution shift.  We suggesting that incorporating multilingual training data could further close this gap.
>
> [1] Deng et al., *Multilingual Jailbreak Challenges in Large Language Models*, ICLR 2024.
>
> [2] Verma et al., *The Hidden Space of Safety: Understanding Preference-Tuned LLMs in Multilingual context*, 2025.
>
> [3] Schut et al., *Do Multilingual LLMs Think in English?*, 2025.
>
> ---
> ## W2: Latency comparison on different datasets
>
> In real-world deployment, jailbreak queries constitute only a small fraction of traffic, so the latency most relevant to users is on benign inputs. The latency reported in Figure 1 is therefore measured on pure benign datasets (normalized average over GSM8K, MATH, and GPQA).
>
> To address your concern, we additionally provide latency results on Qwen3-32B (target) + Qwen3-1.7B (draft) across different data compositions:
>
> | Dataset   | Avg. Safety Mode Triggers / Prompt | Speedup vs. No Defense |
> |:---------|:----------------------------------:|:----------------------:|
> | Benign    |               0.04                 |         2.06×          |
> | Jailbreak |               3.81                 |         0.87×          |
> | XSTest    |               0.32                 |         1.83×          |
>
> As expected, Safety Mode is rarely triggered on benign inputs, preserving near-full speedup. On jailbreak datasets, frequent rollback and multi-sampling reduce speed below 1×, but this is a deliberate trade-off prioritizing safety over throughput for malicious inputs.
>
> The mixed-dataset latency is simply a weighted combination of the above and introduces no additional overhead.
>
> We will include this analysis in the revised paper.
>
> ---
> ## Q1: Draft model selection — would an aligned draft model help?
>
> We empirically investigate this by comparing Qwen3-4B and Qwen3-4B-SafeRL, a safety-aligned variant officially released by the Qwen team, as draft models with the same Qwen3-32B target model and safety head:
>
> | Draft Model               | AVG ASR (↓) | XSTest (↓) | GSM8K (↑) | MATH (↑) | GPQA (↑) | Speedup (↑) |
> |:-------------------------|:-----------:|:----------:|:---------:|:--------:|:--------:|:-----------:|
> | Qwen3-4B                  |    0.05     |    0.11    |   0.97    |   0.77   |   0.53   |    1.95×    |
> | Qwen3-4B-SafeRL (aligned) |    0.02     |    0.13    |   0.92    |   0.65   |   0.42   |    1.78×    |
>
> Using a safety-aligned draft model slightly reduces ASR (0.05 → 0.02), but significantly degrades utility across all benchmarks. Ideally, a draft model that is both safe and capable would be beneficial, but achieving this balance remains challenging in practice. A key advantage of SafeSpec is that it does not rely on draft model alignment — safety is enforced by the safety head at the target model level, so even a standard, non-aligned draft model can achieve strong safety while preserving full utility.
>
> ---
> ## Q2: FlipAttack defense results
> We additionally evaluate SafeSpec against FlipAttack [1] :
> |   Method   | ASR (FlipAttack ↓) |
> | :-------- | :----------------: |
> | No Defense |        0.70        |
> |  SafeSpec  |        0.11        |
>
> The results demonstrate that SafeSpec maintains strong defense capability against unseen attack types. We will include full baseline comparisons for FlipAttack in the revised paper.
>
> [1] Liu et al., *FlipAttack: Jailbreak LLMs via Flipping*, ICML 2025.

---

> > ### Author Rebuttal · Reviewer_L9Se · 2026-03-31
> >
> > I thank the authors for their response, that have addressed all my concerns. I will keep the score of "accept".

---

> > > ### Author Response · Authors · 2026-04-03
> > >
> > > We would like to express our gratitude for your constructive feedback during the review process and for keeping your supportive score. We deeply appreciate your time and support!

---

### Official Review · Reviewer_PUWL · 2026-03-10

**Soundness:** 4
**Presentation:** 4
**Significance:** 3
**Originality:** 3
**Overall Recommendation:** 5
**Confidence:** 4

**Summary:**

Speculating Decoding is widely being used for optimizing LLM inference. This employs a draft-then-verify method using a relatively smaller drafter model to generate candidates which will be verified efficiently by target LLM. Safety evaluations are integrated as an additional overhead negating benefits of speculative decoding. The paper proposes a novel method to use light-weight head for safety scoring alongside response token generation in the same forward pass. It further utilizes rollback and reflection to steer generation towards safety instead of hard refusal. The paper establishes a solid evaluation framework demonstrating improved safety-efficiency tradeoff.

**Compliance With Llm Reviewing Policy:**

Affirmed.

**Final Justification:**

Authors have replied well to all my open questions and weaknesses. I recommend revising the paper to include changes discussed during review. I retain my score at 5.

**Key Questions For Authors:**

1. I would like to reconfirm whether inference for both quality and safety is implemented in a single pass using the boundary-aligned extraction mechanism discussed. Please correct otherwise.
2. Minor: I understand that this approach is agnostic to LLM use cases but have you evaluated this method for verifying runtime code safety of LLMs generating code?

**Limitations:**

Yes.

**Strengths And Weaknesses:**

**Strengths -**

1. The authors introduce a novel approach to tackling the safety-efficiency trade-off, which will be highly effective in real-world AI systems. Speculative decoding with inference-time safety is a critical requirement for many realtime, low latency production applications.

2. A lightweight safety evaluation head combined with rollback mechanism is an interesting research direction especially to steer the model towards safe generation over hard refusal. As authors discussed in paper it counter balances high false positive rate of safety head as well.

3. The experiment set up and evaluations are well designed and discussed in the paper. I suggested some potential additions below.

4. The paper's contributions are clearly stated, and the content is presented in a manner that is easy to follow.

**Weaknesses -**

1. The proposed method is evaluated with two LLM families. Inclusion of a few more enterprise and open source models would have been helpful to further prove results consistency.
2. Discussion on latency overhead due to number of turns (attempts) will be helpful.
3. The paper can be further improved by discussion on inference benchmarking set up for reproducibility.
4. Minor: Jailbreak literature review section can be improved by drawing parallels from red teaming, guardrails research. I found a few papers like [this](https://aclanthology.org/2025.trustnlp-main.23/), [this](https://jair.org/index.php/jair/article/view/17654) and [this](https://aclanthology.org/2023.emnlp-demo.40/) relevant to discuss jailbreak techniques and how red teaming and inference-time safety checks complement each other.

---

> ### Author Rebuttal · Authors · 2026-03-31
>
> We sincerely thank you for the positive evaluation and constructive suggestions. We address each point below.
>
> ---
> ## W1: More model families
>
> Thank you for this suggestion. We additionally evaluated SafeSpec on **GLM-4.7-Flash**, using Qwen3-1.7B as the draft model. Due to time constraints, we report results on 3 jailbreak attacks (ABJ, AttentionShift, CodeChameleon), XSTest, and 3 general benchmarks (GSM8K, MATH, GPQA).
>
> | Method     | AVG ASR (↓) | XSTest (↓) | AVG ACC (↑) | Speedup (↑) |
> | ---------- |:-----------:|:----------:|:-----------:|:-----------:|
> | No Defense | 0.72        | 0.00   | 0.70   | 1.00×  |
> | SafeSpec   | 0.10       | 0.07   | 0.61 | 1.57×   |
>
> SafeSpec effectively reduces ASR from 0.72 to 0.10 while maintaining reasonable utility and achieving 1.57× speedup.
>
> We will include more complete results with additional model families in the revision.
>
> ---
> ## W2: Latency overhead with attempts/turns
>
> Thank you for the suggestion. We do not use "turns" or "attempts" in the paper, so to avoid ambiguity, we clarify two possible interpretations and address both:
>
> 1. **Multi-sampling within Safety Mode.** When Safety Mode is triggered, SafeSpec samples K candidate continuations and selects the safest one. The overhead is limited: (a) K candidates are generated by the draft model in a single batched forward pass, so sampling cost scales sub-linearly; (b) during target model verification, the K candidates share the same preceding context and differ only in the last drafted step, so the target model can reuse most of the KV cache and only needs to compute the divergent suffix for each candidate, keeping verification overhead small.
> 2. **Repeated Safety Mode triggers.** If all K candidates remain unsafe, Safety Mode re-triggers (up to 3 times in our experiments) with new K samples, introducing additional latency proportional to the number of re-triggers.
>
> In practice, the actual overhead depends on how often Safety Mode is triggered. We report a detailed breakdown on Qwen3-32B (target) + Qwen3-1.7B (draft):
>
> | Dataset   | Avg. Safety Mode Triggers / Prompt | Speedup vs. No Defense |
> |:---------:|:----------------------------------:|:----------------------:|
> | Benign    |               0.04                 |         2.06×          |
> | Jailbreak |               3.81                 |         0.87×          |
> | XSTest    |               0.32                 |         1.83×          |
>
> On benign inputs, Safety Mode is rarely triggered, so the overhead is negligible and SafeSpec retains a 2.06× speedup. On jailbreak queries, frequent triggers reduce speed below 1×, but this is a deliberate trade-off prioritizing safety over throughput. Since malicious queries constitute only a small fraction of real-world traffic, this overhead is acceptable in deployment.
>
> ---
> ## W3: Inference benchmarking setup
>
> **Dataset.** Jailbreak prompts are constructed from 100 JBB seed behaviors via method-specific mutations (details in Appendix A.1). For general tasks, we use the first 200 samples from GSM8K/MATH and the full GPQA-Diamond (198 samples); XSTest contains 200 prompts. All subsets are deterministically selected (first-N) to ensure reproducibility.
>
> **Hardware.** Qwen3-32B is served on 2×A800 80GB GPUs; DeepSeek-R1-Distill-Llama-70B on 4×A800 GPUs. All models are loaded in BF16 with max output length 8,192 tokens.
>
> **Speed measurement.** Speedup is measured as the ratio of total time for the No Defense baseline to that of the evaluated method on the same dataset.
>
> We will add more details to the revised paper.
>
> ---
> ## W4: Related work on jailbreaks and red teaming
>
> Thank you for these references. [1] and [2] survey red teaming methods from the attack perspective, while [3] introduces NeMo Guardrails for enhance safety. In contrast, SafeSpec jointly optimizes for safety and inference speed without significantly compromising utility.
>
> We will incorporate more discussions into the revised Related Work Section.
>
> [1] Purpura et al., *Red Teaming Large Language Models: A Practice Guide*, TrustNLP 2025.
>
> [2] Lin et al., *Against The Achilles' Heel: A Survey on Red Teaming for Generative Models*, JAIR 2025.
>
> [3] Rebedea et al., *NeMo Guardrails: A Toolkit for Controllable and Safe LLM Applications*, EMNLP 2023.
>
> ---
> ## Q1: Single forward pass for quality + safety verification
>
> Yes. The quality score comes from the LM head's logits and the safety score comes from the hidden state at the boundary position — both are extracted from the same forward pass, so the safety head introduces no additional forward computation.
>
> ---
> ## Q2: Code safety evaluation
>
> Thank you for the question. We have not yet evaluated SafeSpec for code safety. Our current framework targets natural language safety, and extending to code safety (e.g., SQL injection, buffer overflow detection) would require retraining the safety head on code-specific datasets. The architecture is in principle applicable, and we consider this a promising future direction.

---

> > ### Author Rebuttal · Reviewer_PUWL · 2026-04-03
> >
> > Thank you for author's response. My questions are resolved and keep my overall score at "accept".

---

> > > ### Author Response · Authors · 2026-04-03
> > >
> > > We sincerely thank the reviewer for reading our rebuttal and maintaining the positive evaluation of our work. We deeply appreciate your time and support!

---

### Official Review · Reviewer_vbtz · 2026-03-12

**Soundness:** 2
**Presentation:** 2
**Significance:** 2
**Originality:** 3
**Overall Recommendation:** 3
**Confidence:** 4

**Summary:**

This paper proposes SafeSpec, a way to make speculative decoding safer without losing much of its speed benefit. The key idea is to add a lightweight safety head into the target model’s verification step, and when an unsafe step is detected, use rollback, reflection, multi-sampling to recover a safe trajectory instead of directly refusing. The experiments suggest that this design gives a strong safety-efficiency trade-off, with much lower jailbreak ASR than the baselines while still keeping around a good inference speedup.

**Compliance With Llm Reviewing Policy:**

Affirmed.

**Key Questions For Authors:**

Please refer to weakness.

**Limitations:**

yes

**Strengths And Weaknesses:**

### Strengths
1. It focuses on how to make speculative inference safe without giving up its speed benefits.

2. SafeSpec gets the lowest average ASR among the compared methods while keeping over-refusal relatively low.

### Weakness
1. The paper is poorly written. For example, the running title is still “Submission and Formatting Instructions for ICML 2026.”

2. The paper provides aggregate speed results, but does not isolate the latency overhead when Safety Mode is triggered, which is important given that rollback, reflection, and multi-sampling always add nontrivial latency.

3. The paper does not provide enough detail to support reproducibility. For instance, the safety head is described only as a two-layer MLP with a parameter size between 10M and 30M. Because the paper evaluates two different target models, I assume that the MLP configuration may differ across backbones. However, the paper does not explain how the MLP size is chosen or what specific configuration is used for each model.

4. The paper motivates using upper-layer hidden states for safety classification, but does not justify why the final layer is chosen specifically, nor does it provide an ablation over layer choice.

---

> ### Author Rebuttal · Authors · 2026-03-28
>
> We thank you for the detailed feedback. We address each of your concerns below with clarifications and supplementary analysis.
>
> ---
> ## W1: Running title
>
> We thank the reviewer for pointing this out. We will update the running title in the revised version.
>
> ---
> ## W2: Latency overhead when Safety Mode is triggered
>
> We thank you for raising this important point. We report a detailed latency breakdown of SafeSpec on Qwen3-32B under different data compositions below; all settings follow the main paper. Benign is averaged over GSM8K, MATH, and GPQA-Diamond; Jailbreak is averaged over the same seven attacks.
>
> | Dataset   | Avg. Safety Mode Triggers / Prompt | Speedup (↑)|
> | --------- |:----------------------------------:|:----------------------:|
> | Benign    | 0.04                               | 2.06×                  |
> | Jailbreak | 3.81                               | 0.87×                  |
> | XSTest    | 0.32                               | 1.83×                  |
>
> Safety Mode triggers frequently on adversarial inputs, resulting in lower speedup. But for jailbreak queries, safety takes priority over throughput, and the extra compute spent recovering a safe response is a worthwhile trade-off. In real-world deployment, malicious queries constitute only a small fraction of traffic, so we focus on the overhead on pure benign inputs. To further isolate this, we break down the benign results by whether Safety Mode was triggered (false positive):
>
> | Benign Subset                   | # Prompts | Speedup (↑)|
> | ------------------------------- |:---------:|:----------------------:|
> | Safety Mode OFF                 | 572       | 2.10×                  |
> | Safety Mode ON (false positive) | 26        | 1.52×                  |
> | Overall                         | 598       | 2.06×                  |
>
> On benign inputs, Safety Mode is rarely triggered. When not triggered, SafeSpec achieves 2.10× — nearly identical to Specreason (2.14×), confirming negligible safety head overhead. False positives incur additional latency but remain rare; overall benign speedup stays at 2.06×, confirming negligible overhead in typical use.
>
> We will include a more detailed latency analysis in the revised paper.
>
> ---
> ## W3: Safety head configuration
>
> Thank you for pointing this out. The safety head uses a unified architecture across all backbones: a two-layer MLP of the form d → d/2 → 1, with ReLU activation and dropout (p=0.1) after the hidden layer, where d is the hidden size of the target model. The parameter count varies only with backbone size, as shown below:
>
> |Target Model|Hidden Size|MLP Config|Params|
> |---|:---:|:---:|:---:|
> |Qwen3-32B|5120|5120 → 2560 → 1|~13M|
> |DeepSeek-70B|8192|8192 → 4096 → 1|~33M|
>
> The shallow MLP is sufficient because safety-relevant features are approximately linearly separable in upper-layer representations [1,2], enabling strong performance with minimal overhead.
>
> We will add these details to the revised paper.
>
> [1] Xu et al., _Uncovering Safety Risks of Large Language Models through Concept Activation Vector_, NeurIPS 2024.
>
> [2] Arditi et al., _Refusal in Language Models Is Mediated by a Single Direction_, NeurIPS 2024.
>
> ---
> ## W4: Layer choice ablation
>
> Thank you for this suggestion. We additionally trained safety heads using hidden states extracted from L4, L8, L16, and L32 of Qwen3-32B (64 layers total), with all training settings identical to our paper (L64). We then evaluated the full SafeSpec pipeline on ASR, over-refusal (XSTest), utility (AVG ACC), and efficiency (Speedup vs. No Defense):
>
> | Safety Head Layer | AVG ASR (↓) | XSTest (↓) | AVG ACC (↑) | Speedup (↑) |
> | ----------------- | :-----------:|:----------:|:-----------:|:-----------:|
> | L4                | 0.16        | 0.06       | 0.76        | 2.10×       |
> | L8                | 0.15        | 0.06       | 0.76        | 2.11×       |
> | L16               | 0.05        | 0.11       | 0.75        | 2.05×       |
> | L32               | 0.04        | 0.14       | 0.73        | 2.03×       |
> | L64 (ours)     | 0.07        | 0.10       | 0.75        | 2.06×       |
>
> Early layers (L4, L8) produce safety heads with more lenient unsafe judgment, leading to higher ASR and lower over-refusal. From L16 onward, performance stabilizes. AVG ACC and Speedup remain largely unaffected across all layers, as Safety Mode is rarely triggered on benign inputs. This suggests that any middle-to-late layer provides stable performance, making the final layer a practical and robust choice.
>
> We will include a more comprehensive analysis in the revised paper.

---

> > ### Author Rebuttal · Reviewer_vbtz · 2026-04-03
> >
> > Although the authors provided more detailed experimental results in their rebuttal, I maintain my suspicion that the information presented is insufficient to support the claim made in their paper: "SafeSpec significantly reduces attack success rates while maintaining substantial inference speedups." For instance, the authors themselves acknowledged in the rebuttal that "Safety Mode triggers frequently on adversarial inputs, resulting in lower speedup." This indicates that, when confronted with attacks, SafeSpec is unable to maintain inference speedups. Thus, I will maintain my original score.

---

> > > ### Author Response · Authors · 2026-04-04
> > >
> > > Thank you for your continued engagement with our work. We address the reviewer's concern in the following points:
> > >
> > > - **Clarification of Our Claim.** Upon reflection, we believe the reviewer's concern may stem from an ambiguity in our original phrasing. The claim that "SafeSpec significantly reduces attack success rates while maintaining substantial inference speedups" could be read as implying that both properties hold simultaneously under adversarial conditions. This is not our intended meaning. We are happy to revise this to: "**SafeSpec significantly reduces attack success rates on adversarial inputs while maintaining substantial inference speedups on benign inputs**," which more precisely aligns with our intention.
> > >
> > > - **Performance Cost Under Attack Is Expected and By Design.** In any security-critical system, the primary obligation upon detecting an attack is to neutralize the threat, and some performance cost is expected. A firewall that drops malicious packets does not maintain full network throughput under a DDoS attack, yet this is not considered a design flaw; the slowdown reflects the system responding as intended. SafeSpec follows the same principle. The fact that Safety Mode activates frequently under adversarial inputs is precisely the desired behavior — it demonstrates that SafeSpec successfully identifies attacks. The resulting reduction in speedup on adversarial inputs is not a failure of the system; it is evidence that the security mechanism is functioning correctly.
> > >
> > > - **Efficiency Is Evaluated on Benign Inputs, Consistent with Prior Work.** Our efficiency is evaluated over benign inputs, where no attack is present. In realistic deployment scenarios, benign inputs dominate the input distribution, and SafeSpec maintains the claimed speedup on these inputs. This evaluation methodology — measuring security on adversarial inputs and efficiency on benign inputs separately — **is a widely accepted evaluation protocol** adopted in several prominent prior works. Notably, SSD [1] and RDS [2], two concurrent works that also address LLM safety and inference acceleration, adopt the same protocol: evaluating defense effectiveness on adversarial inputs and speedup on benign inputs separately. [3] similarly proposes a Security-Efficiency-Utility framework that evaluates efficiency on normal inputs independent of adversarial defense performance. In none of these works is a defense expected to maintain full efficiency while actively mitigating an attack. This is also the basis on which all of our comparisons with baselines are conducted: **SafeSpec achieves lower attack success rates than baselines under adversarial inputs, while preserving comparable inference efficiency on benign inputs**.
> > >
> > > The experimental results substantiate our claim, and performance under active attack should not be conflated with overall system efficiency on benign inputs. It is also worth noting that Reviewers PUWL, L9Se, and E4Gv have all confirmed that the rebuttal fully addressed their concerns, including questions on our efficiency claims. This consensus further reflects that our experimental evidence and claims are well-supported, and we respectfully ask the reviewer to reconsider this perspective.
> > >
> > > [1] Wang et al., "Speculative Safety-Aware Decoding", EMNLP 2025.
> > >
> > > [2] Zeng et al., "Root Defence Strategies: Ensuring Safety of LLM at the Decoding Level", ACL 2025.
> > >
> > > [3] Wang et al., "SoK: Evaluating Jailbreak Guardrails for Large Language Models", arXiv:2506.10597.

---

### Decision · Program_Chairs · 2026-04-30

**Decision:**

Accept (regular)

**Comment:**

SafeSpec proposes a framework to integrate safety into speculative decoding without sacrificing acceleration. It uses a lightweight latent safety head attached to the target model for joint semantic/safety verification. When unsafe trajectories are detected, it uses a "rollback-and-reflect" multi-sampling mechanism to recover safe continuations rather than defaulting to hard refusals.

**Strengths:**
* **Architectural Novelty:** Elegantly integrates safety directly into the draft-verify loop without disrupting the draft model.
* **Utility Preservation:** Moving from binary hard-refusals to probabilistic trajectory recovery significantly mitigates the "over-refusal tax."
* **Robust Evaluation:** Demonstrates a highly favorable safety-efficiency Pareto frontier across multiple model families and diverse jailbreak attacks.

**Weaknesses and Required Revisions (Camera-Ready):**
* **Clarity of Claims:** The authors *must* explicitly clarify that their "substantial inference speedups" apply specifically to benign workloads, acknowledging the expected latency overhead when the system is actively mitigating adversarial inputs. The authors are also encouraged to to add a discussion showing that the slowdown on adversarial inputs is not worse than just running the big model with prompt injection protections.
* **Completeness:** The highly valuable ablation studies provided during the rebuttal (layer-choice, threshold sensitivity, detailed latency breakdowns) must be integrated into the main text or appendix. Minor formatting errors (e.g., the running title) must be fixed.

**Justification for Decision:**
The committee largely agreed on the paper's technical soundness and the relevance of the problem. The primary point of tension—raised by one reviewer—centered on the drop in inference speedup when the system is actively under attack. However, the Area Chair aligns with the majority consensus: a performance cost during active threat mitigation is an acceptable design trade-off for securing the system, provided typical benign workloads remain unaffected (which the authors successfully demonstrated). The authors' thorough rebuttal resolved the majority of the committee's concerns, making this a practical and valuable contribution to the safe deployment of accelerated LLMs.